# Novel ST1926 Nanoparticle Drug Formulation Enhances Drug Therapeutic Efficiency in Colorectal Cancer Xenografted Mice

**DOI:** 10.3390/nano14171380

**Published:** 2024-08-23

**Authors:** Sara Assi, Berthe Hayar, Claudio Pisano, Nadine Darwiche, Walid Saad

**Affiliations:** 1Biomedical Engineering Program, American University of Beirut, Beirut 1107 2020, Lebanon; saa124@mail.aub.edu; 2Department of Biochemistry & Molecular Genetics, American University of Beirut, Beirut 1107 2020, Lebanon; bh48@aub.edu.lb; 3Biogem, Institute of Molecular Biology and Genetics, Via Camporeale, 83031 Ariano Irpino, AV, Italy; claudio.pisano@biogem.it; 4Department of Chemical Engineering and Advanced Energy, American University of Beirut, Beirut 1107 2020, Lebanon

**Keywords:** colorectal cancer, retinoids, ST1926, nanoparticles, flash nanoprecipitation

## Abstract

Cancer is a major public health problem that ranks as the second leading cause of death. Anti-cancer drug development presents with various hurdles faced throughout the process. Nanoparticle (NP) formulations have emerged as a promising strategy for enhancing drug delivery efficiency, improving stability, and reducing drug toxicity. Previous studies have shown that the adamantyl retinoid ST1926 displays potent anti-tumor activities in several types of tumors, particularly in colorectal cancer (CRC). However, phase I clinical trials in cancer patients using ST1926 are halted due to its low bioavailability. In this manuscript, we developed ST1926-NPs using flash nanoprecipitation with polystyrene-b-poly (ethyleneoxide) as an amphiphilic stabilizer and cholesterol as a co-stabilizer. Dynamic light scattering revealed that the resulting ST1926-NPs Contin diameter was 97 nm, with a polydispersity index of 0.206. Using cell viability, cell cycle analysis, and cell death assays, we showed that ST1926-NP exhibited potent anti-tumor activities in human CRC HCT116 cells. In a CRC xenograft model, mice treated with ST1926-NP exhibited significantly lowered tumor volumes compared to controls at low drug concentrations and enhanced the delivery of ST1926 to the tumors. These findings highlight the potential of ST1926-NPs in attenuating CRC tumor growth, facilitating its further development in clinical settings.

## 1. Introduction

Nanotechnology is a promising approach for cancer prevention, diagnosis, and treatment, employing advanced targeting, imaging, and delivery techniques [1,2]. This innovative approach has the potential to overcome limitations associated with conventional drug therapy [3]. Nanoparticles (NPs), ranging in size from one to a few hundred nanometers, serve as the fundamental building blocks of nanotechnology. With unique properties such as a high surface area, nanoscale size, and biocompatibility [4], NPs function as efficient drug delivery systems in various tumor models [5,6,7,8,9]. Serving as drug carriers, NPs offer many advantages, including an improved stability and half-life, improved transport, and a prolonged circulation time. Moreover, they utilize the process of the enhanced permeation and retention (EPR) effect, where NP-based drug systems leak preferentially into the tumor site through the permeable vasculature and are retained in the tumor bed due to poor lymphatic drainage [10]. In addition to improving the solubility and bioavailability of existing drugs, NPs can be formulated to selectively release their payload at specific target sites and in response to stimuli [11]. The Food and Drug Administration (FDA USA) approved nanotherapeutic drugs, like Doxil and Abraxane, highlighting their clinical success [12].

Cancer continues to be the second leading cause of death worldwide, resulting in a significant health burden [13]. Colorectal cancer (CRC) is among the most common malignant tumors, as it contributes to more than 1.1 million new cases and 500,000 deaths annually [14]. This cancer ranks third in terms of cancer incidence and second in terms of mortality worldwide [15]. Colorectal cancer treatment involves multiple modalities, including surgery, radiation, and chemotherapy [16], yet the need for safer remedies persists.

In recent years, retinoids have emerged as tumor suppressive agents in multiple cancer models, including breast, melanoma, prostate, and CRC [17]. Retinoids are commonly used in dermatology and leukemia management [18,19,20]. This class of chemical compounds comprising both natural and synthetic derivatives with vitamin A activity, includes all-*trans* retinoic acid (ATRA), widely used to treat acute promyelocytic leukemia [21]. However, ATRA efficacy is limited by poor aqueous solubility and reduced half-life in plasma [22]. To overcome these limitations, synthetic retinoids like ST1926 have been developed [23]. ST1926 displays potent apoptotic activities in various solid malignancies, including ovarian carcinoma [24], rhabdomyosarcoma [25], CRC [26], and hematological malignancies including acute myeloid leukemia [27] and adult T-cell leukemia [28].

We have previously investigated the mechanism of action of ST1926 (Figure 1A) in CRC models [26]. We have demonstrated that ST1926 inhibits the proliferation of human CRC cell lines while sparing normal colon-like cells and significantly reduces the tumor burden in xenograft CRC models. ST1926 induces early DNA damage, S-phase arrest, and apoptosis independently of p53 and p21 status [26]. ST1926 reached phase I clinical trials for patients with advanced ovarian carcinoma; however, its development was halted due to rapid excretion via glucuroconjugation, which resulted in a rapid decline in its plasma concentrations [29].

To improve the therapeutic efficiency of ST1926, we utilized a rapid precipitation approach, termed flash nanoprecipitation (FNP) (Figure 1B), to prepare ST1926-NPs. This process offers several advantages compared to other methods, including continuous processing, an ease of scaling up, and the ability to accommodate multiple active components in the NPs. In addition, from a pharmaceutical manufacturing perspective, FNP presents a convenient technology for the commercial-scale production of NP-based formulations. FNP involves the rapid mixing of solvent and anti-solvent streams in the order of milliseconds [30]. We used a multi-inlet vortex mixer (MIVM) to formulate kinetically controlled NPs. The MIVM consists of four streams; stream 1 contains the hydrophobic drug (ST1926), polystyrene-b-poly (ethyleneoxide) (PS-PEO), and cholesterol, stream 2 is disconnected, and streams 3 and 4 contain distilled water (Figure 1B). This study aims to formulate ST1926-NPs (Figure 1C) with a well-defined particle size and distribution and evaluate their efficacy in CRC therapy both in vitro and in vivo. This research highlights the potential of ST1926 in CRC therapy and calls for its further clinical development.

## 2. Materials and Methods

### 2.1. Materials

ST1926 (E-3-(40-hydroxy-30-adamantylbiphenyl-4-yl) acrylic acid) was purchased from MedChemExpress (Monmouth Junction, NJ, USA) and obtained from Biogem Institute (Ariano Irpino, AV, Italy). PS-PEO diblock copolymer with a polystyrene block size of 1600 g/mol and polyethylene block size of 2900 g/mole was purchased from Polymer source (Dorval, QC, Canada). Water, methanol, acetonitrile, trimethylamine (TEA), and dimethylformamide (DMF; 99.5%) were purchased from Thermo fisher scientific (Waltham, MA, USA). Deionized water was obtained from a Milli-Q purification system.

### 2.2. Formulation of ST1926-Nanoparticles

ST1926-NPs were prepared by FNP with a MIVM mixer. ST1926, PS-PEO, and cholesterol at a mass ratio of 1/2/0.2, respectively, were dissolved in 3 mL of DMF. One of the syringes (stream 1) contained a DMF solution with ST1926, PS-PEO, and cholesterol, while the other two syringes (streams 3 and 4) contained deionized water serving as an anti-solvent (Figure 1B). To formulate NPs, the DMF solution of ST1926, PS-PEO, and cholesterol was mixed using an MIVM chamber, resulting in an output comprising 1:9 DMF/ H_2_O by volume. Using Harvard apparatus PHD2000 syringe pumps, the flow rates of the DMF and water stream used were 12 and 108 mL/min, respectively. The NP solution (10 mL) was collected in deionized water (40 mL). Subsequently, the NP suspensions underwent filtration using hollow fiber filter modules, resulting in a 2 mL sample of concentrated NPs (MIDIKROS, 41.5 CM, Molecular weight cut off: 50 kD, Repligen, Waltham, MA, USA).

### 2.3. Characterization of ST1926-Nanoparticles

The size of the NPs was determined using dynamic light scattering (DLS; Nanoplus HD zeta/nanoparticle analyzer, Particulate systems) following NP formation and after 24 h at 4 °C. The particle size distribution was determined by measuring light scattering at a fixed angle of 90° at 25 °C. The CONTIN algorithm was employed to obtain the number size distribution of NP suspensions. The morphology of the NP formulation was observed using a scanning electron microscope (MIRA 3 LMU, Tescan, Brno, Czech Republic) with an acceleration voltage of 10 kV. One drop of the NP suspension was placed on a coverslip on a carbon-coated copper grid and dried at room temperature overnight. The drug loading efficiency (DLE) and drug loading content (DLC) of ST1926-NPs were calculated according to Equations (1) and (2), respectively.
(1)$$\mathrm{DLE}\;\left(\%\right)=\frac{Amount\;of\;drug\;in\;NPs}{Total\;amount\;of\;drug\;used\;in\;formulation\;processing}\times 100$$
(2)$$\mathrm{DLC}\;\left(\%\right)=\frac{Amount\;of\;drug\;in\;NPs}{Weight\;of\;the\;NP\;formulation}\times 100$$

### 2.4. High-Performance Liquid Chromatography

The quantification of ST1926 in NP suspensions was performed using high-performance liquid chromatography (HPLC). A sample of ST1926-NP was dissolved in DMF and vortexed to disintegrate the NP and obtain dissolved ST1926. HPLC analysis was performed using an AGILENT 1260 Infinity II model equipped with a quaternary pump, degasser, autosampler, thermostated column compartment, and UV-Vis detector. The column used for analysis was a Kinetex EVO C18 column of a 5 µm particle size, 4.6 mm internal diameter, and 250 mm length, accompanied by a guard column (2.1 × 4.6 mm) (Phenomenex, Torrance, CA, USA). The column flow rate was set at 0.5 mL/min, and detection was carried out at a wavelength of 320 nm. The mobile phases consisted of phase A (0.05% TEA in water) and phase B (0.05% TEA in methanol) (Appendix A, Appendix A). The elution is gradient, starting with an initial composition of 50% mobile phase B and a linear increase to 99% mobile phase B over 2 min, followed by constant conditions maintained until 12 min.

### 2.5. Cell Culture

The human CRC cell line HCT116 was obtained from the American Tissue Culture Collection, ATCC, Manassas, VA, USA. HCT116 cells (passages between 18 and 22) were cultured in RPMI 1640 (Lonza, Basel, Switzerland) medium supplemented with 10% fetal bovine serum (Sigma-Aldrich, St. Louis, MO, USA)., 100 U/mL of penicillin-streptomycin antibiotics (Lonza, Basel, Switzerland), and 1 mM of sodium pyruvate solution (Lonza, Basel, Switzerland). The NCM460 normal-like colon cell line (INCELL Corporation, LLC, San Antonio, TX, USA) was derived from normal colon epithelium and was maintained in M3:Base medium (INCELL Corporation, LLC, San Antonio, TX, USA) supplemented with 10% fetal bovine serum. Cells were incubated at 37 °C under a humidified (95% air, 5% CO_2_) atmosphere.

### 2.6. Cell Growth Assay

Cell growth was assessed using thiazolyl blue tetrazolium bromide (MTT) dye (Sigma, St. Louis, MO, USA) according to the manufacturer’s instructions. HCT116 cells were seeded in 96-well plates at a density of 5 × 10^3^ cells per well in triplicate and cultured for 24 h. Subsequently, the cells were treated with different concentrations of ST1926 or ST1926-NPs, along with their respective controls, for different time intervals: 24 h, 48 h, and 72 h. An enzyme-linked immunosorbent assay (ELISA) microplate reader (Multiscan Ex, Thermo Electron Corporation, US) was used to measure the absorbance at 595 nm.

### 2.7. Cell Viability Assay

Cell viability was determined using the trypan blue dye exclusion assay. HCT116 cells were seeded in 24-well plates at a density of 2 × 10^4^ cells per well in triplicate and cultured for 24 h. Subsequently, the cells were treated with different concentrations of ST1926 or ST1926-NPs, along with their respective controls for different time intervals: 12, 24, 48, and 72 h. At each time point, the cells were harvested using trypsin/EDTA. A 50 µL sample was collected from a 500 µL cell suspension and mixed with 50 µL trypan blue. Live and dead cells were then counted using a hemocytometer.

### 2.8. Cell Cycle Analysis

Cell cycle analysis was performed using the propidium iodide (PI) assay. HCT116 cells were seeded in 100 mm cell culture dishes at a density of 5 × 10^5^ per well and cultured for 24 h. The cells were then treated with 0.5 µM of ST1926 or ST1926-NP for different time intervals: 24, 48, and 72 h. At each time point, cellular pellets were obtained and incubated with 100 µL RNase (Roche Diagnostics, Basel, Switzerland) for 1 h, followed by resuspension in up to 500 µL 1 × PBS. Subsequently, the cells were stained with 30 µL PI (Sigma-Aldrich) and incubated for 10 min in the dark. Flow cytometry analysis of 10,000 events was performed using a FACScan flow cytometer (Becton Dickinson, Franklin Lakes, NJ, USA), and cell cycle distribution was verified using BD FACSDIVA software version 8.0.

### 2.9. Western Blotting

Protein lysates were separated by 8–12% sodium dodecyl sulfate polyacrylamide gel electrophoreses (SDS-PAGE) and subsequently transferred to nitrocellulose membranes. The membranes were then blocked with 4% non-fat milk in TBS and incubated with specific primary antibodies: Poly (ADP-ribose) polymerase (PARP) (Santa Cruz, Dallas, TX, USA) (1:1000), phosphorylated H2A histone family member X (γH2AX) (Cell Signaling) (1:1000), and GAPDH (Abnova, Taipei, Taiwan) (1:20,000) at 4 °C overnight. The following day, the membranes were washed at room temperature, followed by incubation with the corresponding secondary antibodies for 1 h. Immunoreactive bands were detected using a ClarityTM western ECL substrate (ECL, Bio-Rad, Hercules, CA, USA) and the ChemidocTM MP imaging System (Bio-Rad).

### 2.10. In Vivo Antitumor Efficacy

The mice protocols were approved by the Institutional Animal Care and Use Committee of the American University of Beirut (Approval code: 23-03-606). NOD-scid IL2rg^null^ (NSG) male and female mice aged 4–6 weeks old, with an average weight of 20 g, were subcutaneously injected with 5 × 10^6^ HCT116 cells. The mice were divided randomly into four groups (Table 1): Control (*n* = 7), ST1926 (*n* = 7), Control NP (*n* = 10), and ST1926-NP treatment (*n* = 10). Seven days post-HCT116 cells injection, the xenografted mice were intraperitoneally injected with 4 mg/kg ST1926-NP every other day for up to 2 weeks. The mice were weighed once a week, and tumor volumes (V) were measured twice weekly using calipers and the formula provided below. The mice were monitored every other day for any changes in fur, movement, etc. and checked for signs of toxicity. Tumors were collected after sacrifice to evaluate ST1926 concentrations.
$$V=\frac{L\times W^{2}}{2}$$

### 2.11. Preparation of Standard Solutions

A 1 mg/mL stock solution of ST1926 was prepared for the assay of plasma and tumor samples. This stock was then diluted in DMF to create working solutions with concentrations ranging from 10 to 10,000 ng/mL. All stock and working solutions were stored in the dark at −20 °C until further use. For assaying plasma or tumor homogenates, calibration standards were prepared by adding 10 µL of different working solutions to 90 µL of either control murine plasma (Appendix A) or control tumor homogenate (Appendix A). This resulted in final standard concentrations of 1, 5, 10, 20, 50, 100, 250, 500, and 1000 ng/mL.

### 2.12. Preparation of Plasma Samples

Plasma samples (100 µL) were mixed with 600 µL of DMF and vortexed vigorously for 1 min. The plasma mixture was centrifuged for 10 min at 15,000 g and 4 °C. The supernatant was then recovered, and 10 µL were injected into the HPLC-Mass Spectrometry/Mass Spectrometry (HPLC-MS/MS) system.

### 2.13. Preparation of Tumor Homogenate Samples

The control and tumor samples were weighed. Cell lysis buffer (500 µL) was added per 100 mg of tissue. The samples were incubated on ice for 30 min to ensure complete lysis. The tissues were then homogenized using an ultra-turrax for 1 min, on average. The homogenates were centrifuged at 16,000× *g* for 10 min at 4 °C, and the supernatants were transferred to new microcentrifuge tubes. An aliquot of the tumor homogenate (100 µL) was mixed with DMF at a 1:6 (*w*/*v*) ratio. The mixture was centrifuged at 16,000× *g* for 10 min at 4 °C. The resulting supernatant was transferred to an HPLC vial, and 10 µL was injected into the HPLC-Quadrupole Time of Flight/Mass Spectrometry (HPLC-QTOF/MS) system.

### 2.14. High-Performance Liquid Chromatography–Mass Spectrometry/Mass Spectrometry Conditions for the Quantification of ST1926 in Plasma Samples

The HPLC (Waters Alliance 2695 separation module) coupled with Micromass Quattro micro-API triple quadrupole M) (Waters) was used to quantify ST1926 in plasma samples. The samples were separated on a chromatographic column XTerra *MS* C18 3.5 µm, 100 mm × 2.1 mm, coupled with a guard column of the same material (Waters). The mobile phases consisted of phase A (0.05% formic acid in water) and phase B (0.05% formic acid in 50% acetonitrile and 50% methanol). The HPLC system operated at a flow rate of 0.4 mL/min under the following gradient settings: from 99% mobile phase A and decreasing to 10% in 2 min, maintaining this step for 6 min, and then returning to 99% mobile phase A and maintaining for an additional 6 min. The triple quadrupole mass spectrometer was operated in the negative ion electrospray ionization mode, with argon used as the collision gas in the collision cell. The cone voltage was set at 50 V. The mass spectrometer operated in the multiple reaction monitoring (MRM) mode, permitting the passage of the [M-H]- ion of ST1926 (*m*/*z* 373.4) through the first quadruple into the collision cell. Following fragmentation, the specific product ions of ST1926 were monitored in the third quadruple at *m*/*z* 329.6 and 271.5.

### 2.15. High-Performance Liquid Chromatography–Quadrupole Time-of-Flight/Mass Spectrometry Conditions for the Quantification of ST1926 in Tumor Samples

The HPLC–QTOF/MS (X500R, SCIEX) was used to quantify ST1926 in the tumor samples. The samples were separated on a chromatographic column Luna Omega Polar C18 3 µm, 100 mm × 2.1 mm coupled with a guard column of the same material. The mobile phases consisted of phase A (0.05% formic acid in water) and phase B (0.05% formic acid in 50% methanol 50% acetonitrile). The chromatographic separation operated at a flow rate of 0.5 mL/min, applying the following gradient steps: 50% mobile phase B, a linear increase to 99% mobile phase B over 2 min, followed by constant conditions maintained until 9 min. The mass spectrometer was operated in negative ion electrospray ionization mode and in MRM mode. The curtain gas and collision-activated dissociation gas were set to 35 and 7 psi, respectively. The declustering potential (DP) was set at −90 V and the collision energy was fixed at −15 V.

### 2.16. Statistical Analysis

All results represent the average of three independent experiments ± the standard error of the mean (SEM), unless indicated otherwise. Statistical significance was determined using GraphPad prism version 9. *p* < 0.05 is considered statistically significant: * *p* < 0.05, ** *p* < 0.01, *** *p* < 0.001.

## 3. Results

### 3.1. Formulation and Characterization of ST1926-Nanoparticles

ST1926 is a highly hydrophobic drug, which undergoes major glucuroconjugation on its phenolic hydroxyl group, affecting its bioavailability. ST1926 was formulated into NPs with a drug-to-polymer-to-co-stabilizer mass ratio of 1:2:0.2, using 5 mg of ST1926, 10 mg of PS-PEO copolymer, and 1 mg of cholesterol. PS-PEO was selected as the amphiphilic stabilizer due to its biocompatibility [31]. Cholesterol was added as a hydrophobic co-stabilizer to facilitate the secure binding of the PS hydrophobic block to the core hydrophobic ST1926 drug, allowing the hydrophilic PEO block to face the external aqueous phase [32]. Drugs within the ACD/Log P (measure of hydrophobicity) range of 2 to 9 often exhibit instability attributed to Ostwald ripening and the non-equilibrium molecular orientation of amphiphilic stabilizers. The incorporation of cholesterol as a co-stabilizer addresses this critical limitation of FNP, considering that the majority of drugs fall within this ACD/Log P range—in particular, the ST1926 value is 6.74.

The size of the ST1926-NPs was investigated using DLS and was found to be, on average, 97 nm, with a polydispersity index of 0.206 (Figure 2A). The intensity-based average diameter was 265 nm (Appendix A). ST1926-NPs were shown to remain stable at 4 °C for 24 h. The DLE and DLC of the ST1926-NPs were determined to be 30% and 7%, respectively (Figure 2B). The morphology of ST1926-NPs was examined by a scanning electron microscope (Figure 2C), revealing a spherical shape. Subsequently, ST1926-loaded polymeric NPs were employed for further in vitro and in vivo testing. We also prepared control-NPs by using PS-PEO and cholesterol at the same concentrations (Appendix A).

### 3.2. ST1926 Naked Drug and Nanoparticle Formulation Exhibit Comparable Potent Growth Inhibitory Effects in Colorectal Cancer Cells

To assess the effect of ST1926-NPs on the viability of CRC cells, we conducted both a colorimetric MTT assay and a trypan blue exclusion assay. The HCT116 cell line was selected, as it is widely used in human CRC studies. We observed comparable cytotoxic effects of ST1926 naked drug and ST1926-NPs on the growth of human HCT116 CRC cells. ST1926-NP concentrations as low as 0.5 µM significantly inhibited the proliferation of the treated cells, which was evident as early as 24 h post-treatment (Figure 3A). The half-maximal inhibitory concentration (IC_50_) at 24 h of the ST1926 naked drug and its NP-formulation was 0.5 µM (Figure 3A and Appendix A). However, the ST1926-NPs were less sensitive when tested on normal-like human colon mucosal epithelial cells NCM460. In fact, 0.5 µM ST1926-NPs did not affect the growth of normal-like colon cells up to 72 h post-treatment (Appendix A).

The MTT results were confirmed using the trypan blue exclusion assay, revealing consistent trends in assessing cell viability. Notably, using the trypan blue exclusion assay, the IC_50_ of ST1926 and ST1926-NPs was determined to be approximately 0.1 µM (Figure 3B).

### 3.3. ST1926 Naked Drug and Nanoparticle Formulation Induce Sub-G_1_ Cell Accumulation and Cell Cycle Arrest

To investigate the mechanisms underlying the growth inhibition caused by ST1926 and its NP formulation, cell cycle analysis was performed in HCT116 cells treated with 0.5 µM of ST1926 and its NP formulation for up to 72 h. The cells treated with ST1926-NP exhibited a significant increase in cell accumulation in the presumably apoptotic sub-G_1_ region as early as 24 h post-treatment compared to its control (Figure 4A). However, the cells treated with ST1926 showed significant sub-G_1_ cell accumulation after 48 h (Figure 4A). Approximately 70% of ST1926-NP treated cells, and 60% of ST1926-treated cells, accumulated in the sub-G_1_ region at 72 h. As for the cycling cells, we observed a significant G_0_/G_1_ cell cycle arrest in the cells treated with ST1926 and ST1926-NP at 24 h post-treatment (Figure 4A).

### 3.4. ST1926 Naked Drug and Nanoparticle Formulation Induce Apoptosis and Early DNA Damage in Colorectal Cancer Cells

We further examined the mechanism of cell death, since HCT116 CRC cells accumulated in the apoptotic sub-G_1_ region upon treatment with ST1926 or its NP formulation. Apoptosis is characterized by the proteolytic cleavage of PARP by caspases and the phosphorylation of the DNA damage indicator H2AX to γH2AX. To determine the effects of ST1926 and ST1926-NP on the levels of PARP and γH2AX, western blotting was performed at a concentration of 0.5 µM. A minor PARP cleavage was detected in HCT116 cells after 12 h of ST1926 or ST1926-NP treatments, with cleavage becoming more prominent after 24 h of treatment with either compound (Figure 4B). Moreover, ST1926 and ST1926-NP increased γ-H2AX as early as 12 h post-treatment, suggesting early DNA damage, which was more prominent at 24 h post-treatment (Figure 4B).

**Figure 3 nanomaterials-14-01380-f003:**
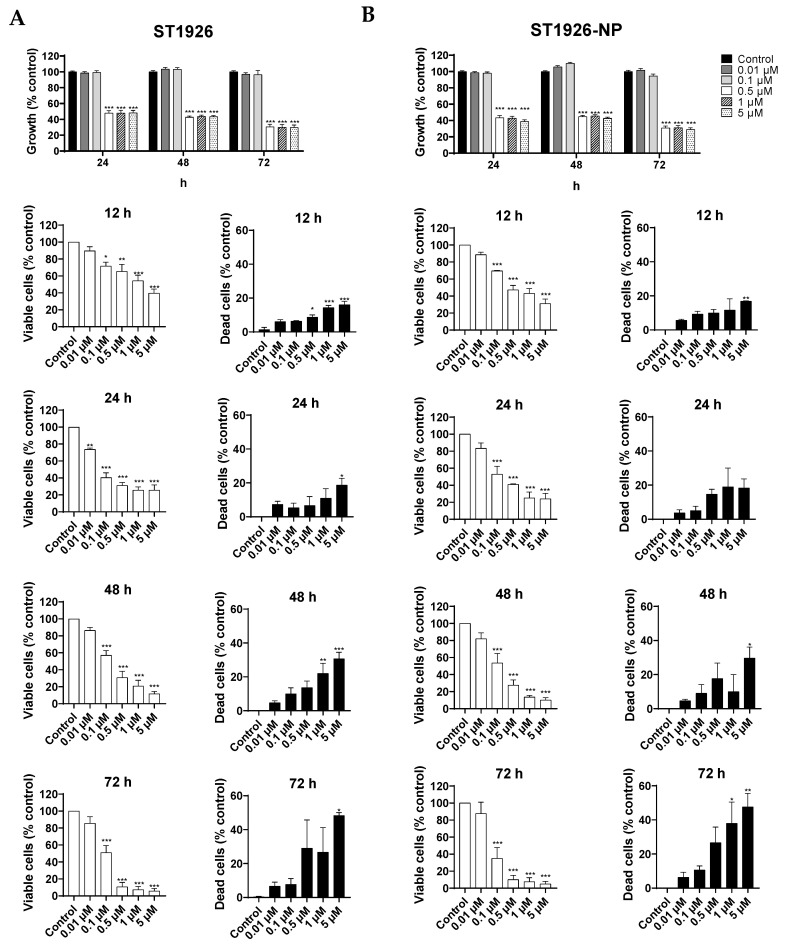
ST1926 and ST1926-nanoparticle (ST1926-NP) treatments inhibit colorectal cancer cell growth and reduce viability. (**A**) Inhibition of colorectal cancer cell growth by ST1926 and ST1926-NP. HCT116 cells were treated with the indicated concentrations of ST1926 or ST1926-NP for up to three days, and cell growth was examined using the MTT colorimetric assay; (**B**) Inhibition of colorectal cancer cell viability by ST1926 and ST1926-NP. Cell viability was examined in triplicate wells using the trypan blue exclusion assay. Results are expressed as the percentage of the control group set as 100% and represent the average of three independent experiments ± SEM. Significance from the control is demonstrated by * *p* < 0.05, ** *p* < 0.01, and *** *p* < 0.001.

**Figure 4 nanomaterials-14-01380-f004:**
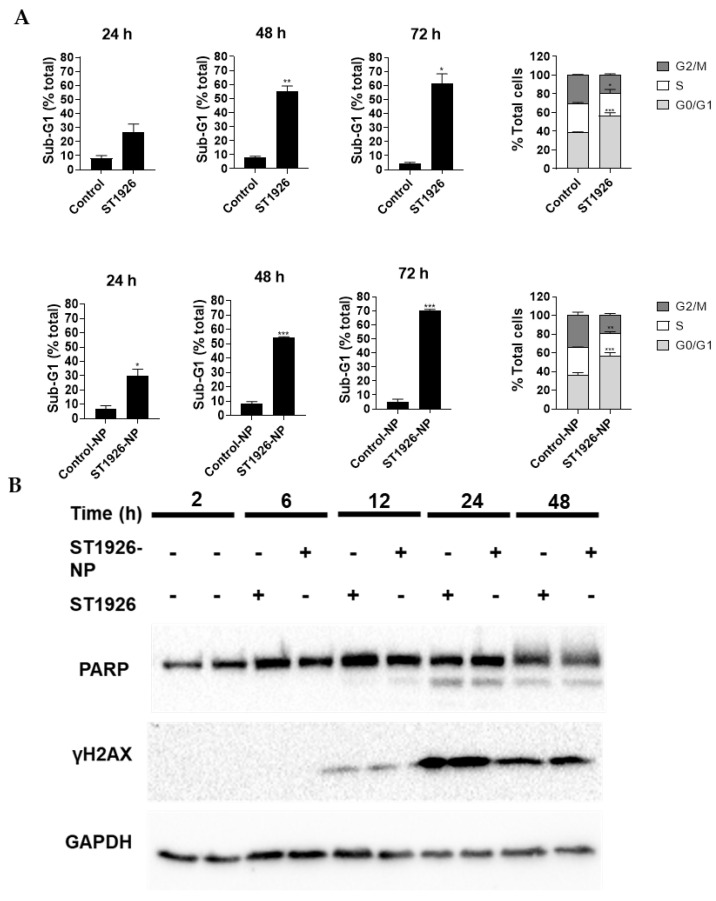
ST1926 and ST1926-nanoparticle (ST1926-NP) treatments increase sub-G_1_ cell accumulation and induce apoptosis and DNA damage in colorectal cancer cells. (**A**) Effects of ST1926 and ST1926-NP on the cell cycle distribution of colorectal cancer cells. HCT116 cells were treated with 0.5 µM of ST1926 or ST1926-NP and their respective controls for up to 72 h. Results represent the average of three independent experiments, (±SEM). Significance from the control is demonstrated by * *p* < 0.05, ** *p* < 0.01, and *** *p* < 0.001; (**B**) Colorectal cancer cells were treated with 0.5 µM of ST1926 or ST1926-NPs. Total SDS protein lysates (50 µg/lane) were immunoblotted against PARP and γH2AX antibodies. This is a representative blot of three independent experiments.

### 3.5. Formulating ST1926 into Nanoparticles Increases the Tberapeutic Efficiency of ST1926 in Treated Mice

We next investigated whether ST1926-NP increases the efficiency of the treatment versus the naked drug in vivo. Numerous studies have previously tested the effects of ST1926 in several in vivo tumor models at concentrations ranging from 15 mg/kg [26] up to 50 mg/kg body weight [23]. These studies, among others, have shown that these concentrations reduce tumor growth in mice and increase animal survival with no detectable signs of toxicity. We opted to test lower concentrations of ST1924 (4 mg/kg) and investigate whether the NP formulation increases the therapeutic efficiency of ST1926-NP versus the naked free drug.

Towards this end, we selected the CRC xenograft mouse model using HCT116 cells subcutaneously injected into the flank of immunocompromised male and female NSG mice. Seven days post-HCT116 cells injection, the HCT116 cells-xenografted mice were intraperitoneally treated with 4 mg/kg ST1926, ST1926-NP, or their respective controls three times a week for up to 16 days. The changes in tumor volumes and body weights throughout the treatment duration are shown in Figure 5A,B, respectively.

As expected, ST1926 at 4 mg/kg did not reduce the tumor volumes (Figure 5A). On the other hand, the average tumor volumes of mice injected with control-NP increased from approximately 30 mm^3^ to 1100 mm^3^, whereas those in mice treated with ST1926-NP increased from approximately 30 mm^3^ to ~700 mm^3^ after 16 days of treatment (Figure 5A). The average change in the tumor volume reduction in mice treated with ST1926-NP compared to its control was significant at day 13 post-treatment (Figure 5A). When the tumor volume of the control mice exceeded 1100 mm^3^, we terminated the experiment for humane reasons. The animal weights of all four groups of mice did not drop significantly over two weeks, indicating that the treatments were well tolerated (Figure 5B). In summary, our results indicate that ST1926-NP enhanced the therapeutic efficiency of ST1926 at 4 mg/kg low concentrations.

### 3.6. Plasma ST1926 Concentrations Did Not Significantly Differ in Mice Treated with Naked or Formulated ST1926

We examined the plasma concentrations of ST1926 following the intraperitoneal administration of 4 mg/kg of ST1926 and ST1926-NP in NSG male and female mice. The initial plasma concentrations at 0.5 h post-treatment were comparable: 310 ng/mL for ST1926 and 315 ng/mL for ST1926-NP. Both concentrations decreased steadily, reaching 76 ng/mL for ST1926 and 20 ng/mL for ST1926-NP at 2 h post-treatment. No significant difference in plasma concentrations was observed between ST1926 and ST1926-NP at any time point (Figure 6).

### 3.7. Formulating ST1926 into Nanoparticles Enhances Drug Delivery to the Tumors

We studied whether ST1926 formulation enhanced drug delivery to the tumors versus the naked drug. ST1926 concentrations in the tumors were analyzed and showed that ST1926 was detected in 6 out of 10 tumors treated with ST1926-NP, with an average concentration of 44.4 ng/mL (Table 2). In contrast, no ST1926 was detected in the tumor samples treated with ST1926 alone, as it was below the limit of detection (LOD) (1 ng/mL) (Table 2).

## 4. Discussion

The success rate of new cancer drugs in clinical development is low and does not exceed 3.4% [33]. The clinical testing of ST1926 for ovarian cancer was halted in Phase I due to rapid glucuronidation at its phenolic hydroxyl group, resulting in its low bioavailability and rapid liver excretion [29]. This observation was supported by findings indicating short-lived plasma concentrations of ST1926 in humans [29] and mice [25]. These challenges emphasize the urgent need for innovative solutions in drug formulation. Given ST1926’s potency and relative safety in several tumor models, significant efforts have been made to develop analogs with improved properties [34,35]. Utilizing NPs to enhance the bioavailability of ST1926 presents a promising platform for improving drug delivery and efficacy, as demonstrated by several studies [36,37,38,39,40].

Colorectal cancer, ranking among the most frequently diagnosed malignant tumors worldwide [14], requires effective treatment options. We have previously demonstrated the promising efficacy of ST1926 in CRC treatment [26]. ST1926 was shown to potently inhibit the proliferation of human CRC cell lines while sparing normal cells and to significantly reduce the tumor burden in a xenografted CRC model. ST1926 induced early DNA damage, S-phase arrest, and apoptosis independently of p53 and p21 status [26]. Alternatively, polymer-stabilized ST1926-NP formulations have been developed previously in our laboratory and have been shown to improve the anti-tumor activities of ST1926-NP at four-fold lower concentrations (7.5 mg/kg) than the naked ST1926 (30 mg/kg) drug in human acute myeloid leukemia models [27].

We further developed ST1926-NPs and examined their effect on in vitro and in vivo CRC models. In this study, we showed that ST1926-NPs inhibited the growth and viability of CRC cells at concentrations as low as 0.5 µmol/L. This anti-tumor effect was comparable to that of the naked drug. The mechanism of action of ST1926-NPs in CRC is similar to that of ST1926, which was reported in tested solid tumors and in hematological malignancies [24,25,27,41]. ST1926 has been identified as a genotoxic drug that induces early DNA damage in various types of tumor cells [25,27]. Here, we show that both ST1926 and ST1926-NP resulted in early DNA damage at sub-micromolar concentrations. The effects of ST1926 were reversed by co-treatment with DNA damage inhibitors, demonstrating the significant involvement of ST1926 in DNA damage [25]. Furthermore, we have shown that DNA polymerase α (POLA1) is a target of ST1926, and its mutation inhibits DNA damage and confers resistance in CRC cells [26].

While in vitro evaluation is useful as an initial step in assessing the therapeutic effectiveness of anticancer drugs, it does not directly correlate with in vivo outcomes. This is due to the complexity of the tumor microenvironment in vivo, such as the influence of blood flow on interactions between peptides and target receptors, clearance by the reticuloendothelial system, and the accumulation of NPs in tumors through the EPR effect [42]. As such, we studied the anti-tumor effects of ST1926-NP versus ST1926 in CRC cells-xenografted mice.

Previous studies have demonstrated the anti-tumor effectiveness of ST1926 across different malignancies, with effective doses ranging from 15 mg/kg [26] to 50 mg/kg [23]. Notably, 4 mg/kg concentrations have not been studied for their efficacy in inhibiting tumor growth. Interestingly, our findings revealed a significant effect in inhibiting tumor growth in mice treated with 4 mg/kg of ST1926-NP, whereas naked ST1926 at 4 mg/kg failed to elicit any detectable effect. This suggests that NP delivery technology enhances therapeutic outcomes with lower dosages.

Understanding the pharmacokinetics of ST1926 is essential for elucidating its mechanism of action. We measured plasma concentrations of ST1926 in both its NP formulation and as a naked drug. Intriguingly, the NP formulation did not enhance the plasma ST1926 concentrations compared to the naked drug. However, plasma concentrations may not fully reflect the drug’s distribution or efficacy. As such, subsequent tumor studies revealed that ST1926-NP accumulated in tumors, while naked ST1926 failed to do so. This suggests that ST1926-NP may have reduced absorption into the bloodstream or rapid clearance but enhanced distribution to extravascular tissues, including tumors. Nanoparticles are capable of being distributed to the tissues directly, where they exert their therapeutic effects [43]. In fact, research has focused on targeting NPs to tumors via the EPR effect [44,45,46].

It would be interesting in future studies to explore the tissue distribution profile of ST1926-NPs versus the naked drug. This may provide insights into the targeting efficacy of ST1926-NPs and any off-target effects. Although the use of animal models is essential in anticancer drug development, these models also have limitations that should be taken into consideration when designing drug formulations [47].

In conclusion, our results highlight the potential of ST1926-NP in CRC therapy, presenting a promising platform for further drug clinical development using formulation techniques. Our findings revealed a significant effect in inhibiting tumor growth in mice treated with 4 mg/kg of ST1926-NP, whereas naked ST1926 at the same dosage failed to elicit any detectable effect. Moreover, our formulation enhanced the delivery of ST1926 into tumors. This demonstrates the superior effectiveness of our NP formulation, making it a superior option compared to the naked drug.

## Figures and Tables

**Figure 1 nanomaterials-14-01380-f001:**
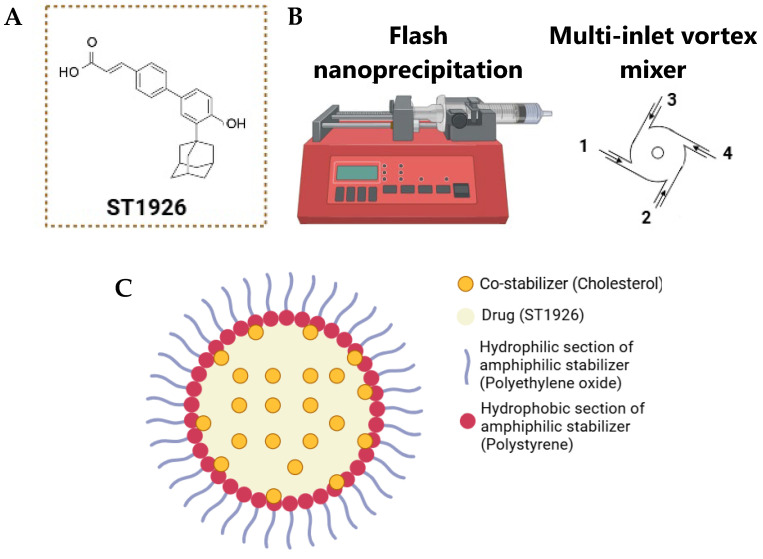
(**A**) Chemical structure of ST1926. (**B**) Schematic illustration of flash nanoprecipitation and multi-inlet vortex mixer. (**C**) Schematic illustration of ST1926-nanoparticle formulation. Stream 1 contains the hydrophobic drug (ST1926), polystyrene-b-poly (ethyleneoxide) (PS-PEO), and cholesterol, stream 2 is disconnected, and streams 3 and 4 contain distilled water.

**Figure 2 nanomaterials-14-01380-f002:**
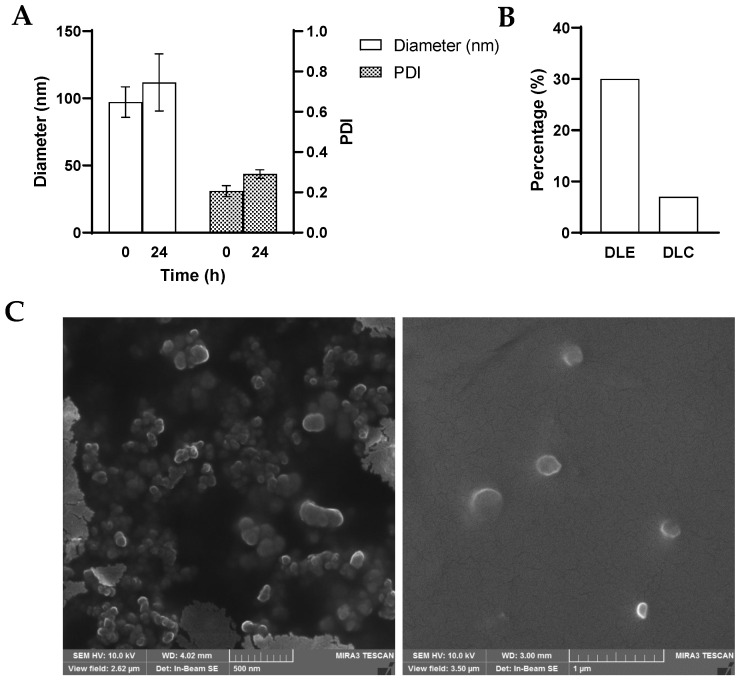
(**A**) Size and polydispersity index (PDI) of ST1926-nanoparticles. The size and PDI of the polymer-coated ST1926 was determined by dynamic light scattering initially and after 24 h at 4 °C. Results represent the average of three independent experiments. (**B**) Drug loading efficiency (DLE) and drug loading content (DLC) of ST1926-nanoparticles. (**C**) Scanning electron microscopy images of ST1926-nanoparticles.

**Figure 5 nanomaterials-14-01380-f005:**
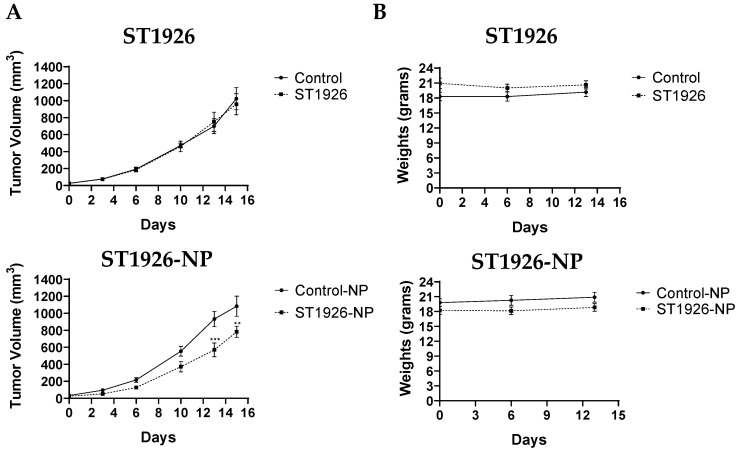
ST1926 nanoparticle formulation reduces the tumor volume in vivo. In total, 5 × 10^6^ HCT116 cells were injected subcutaneously into the flank of male and female NSG mice. After tumor development, the mice were divided into four groups: Control (*n* = 7), ST1926 (*n* = 7), control nanoparticle (NP) (*n* = 10), and ST1926-NP (*n* = 10). The mice were treated intraperitoneally with 4 mg/kg of ST1926, ST1926-NP, or their respective controls three times a week for up to 16 days. (**A**) Average tumor volumes (±SEM) of mice treated with ST1926 and ST1926-NP and their respective controls. ** *p* < 0.01 and *** *p* < 0.001 are considered statistically significant.; (**B**) Average animal weights (±SEM) of mice treated with ST1926 and ST1926-NP and their respective controls.

**Figure 6 nanomaterials-14-01380-f006:**
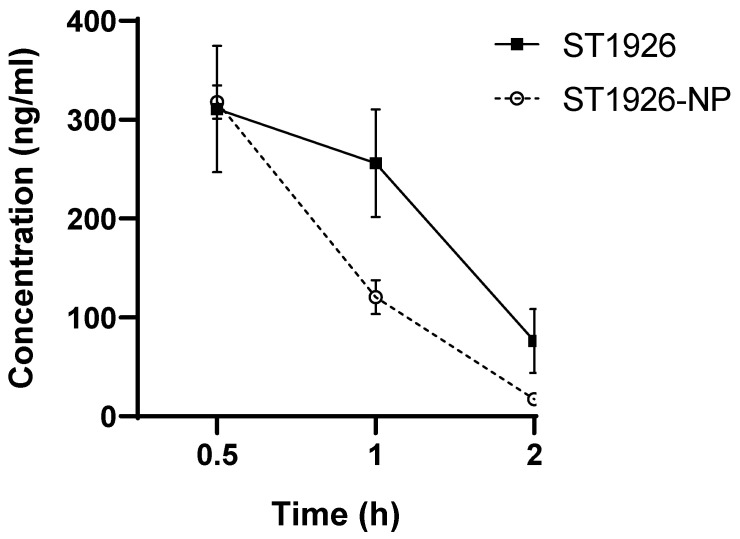
Plasma ST1926 concentrations of ST1926 and ST1926-nanoparticles (NPs)-injected mice. NSG mice were intraperitoneally injected with 4 mg/kg of ST1926 and ST1926-NP (*n* = 3/time point), and plasma ST1926 concentrations (average ± SEM) at the indicated time points were determined by HPLC-MS/MS.

**Table 1 nanomaterials-14-01380-t001:** In vivo experimental mice group distribution.

Treatment	Control	ST1926	Control-NP *	ST1926-NP *
Number of mice	7 (3 males, 4 females)	7 (4 males, 3 females)	10 (5 males, 5 females)	10 (4 males, 6 females)

* NP: Nanoparticles.

**Table 2 nanomaterials-14-01380-t002:** Concentrations of ST1926 in tumor samples.

Sample Name	Calculated Concentrations (ng/mL)
ST1926-1 (female)	<LOD *
ST1926-2 (female)	<LOD *
ST1926-3 (female)	<LOD *
ST1926-4 (male)	<LOD *
ST1926-5 (male)	<LOD *
ST1926-6 (male)	<LOD *
ST1926-7 (male)	<LOD *
ST1926-NP *-1 (female)	6.083
ST1926-NP *-2 (female)	74.84
ST1926-NP *-3 (female)	31.28
ST1926-NP *-4 (female)	43.41
ST1926-NP *-5 (male)	42.3
ST1926-NP *-6 (male)	68.75
ST1926-NP *-7 (female)	<LOD *
ST1926-NP *-8 (female)	<LOD *
ST1926-NP *-9 (male)	<LOD *
ST1926-NP *-10 (male)	<LOD *

* NP: nanoparticle, LOD: limit of detection.

## Data Availability

All data generated or analyzed in this manuscript are included in this article and in Appendix A.

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
