# Peer review of "Novel ST1926 Nanoparticle Drug Formulation Enhances Drug Therapeutic Efficiency in Colorectal Cancer Xenografted Mice"

_nanomaterials, 2024, doi:10.3390/nano14171380_

Round 1

Reviewer 1 Report

Comments and Suggestions for Authors

The authors here report the use the use of NP encapsulated adamantyl retinoid ST1926 for colorectal cancer therapy. My comments are as follows:

1.      Schematic representation must be provided as Figure 1

2.      The methods must be described in detail (for example, the MWCO for the filters used in NP formulation, Cell source, passage number, age of mice etc.)

3.      Key data like encapsulation efficiency of the drug and its loading content are missing. The DLS plots based on intensity and number should be provided. Zeta potential is a key factor in NP stability, this data is missing.

4.      What is the critical aggregation constant (CAC) of the NPs?

5.      The authors should provide a drug release plot of the NPs.

6.      Do the authors have TEM images? SEM images are not very clear and since the surface morphology is not being investigated, it is more useful to have TEM images as well as a histogram representing the size distribution.

7.      The authors have tested only one cell line, this is insufficient to draw conclusions. They should test their formulations and relevant controls on a non-cancerous cell line and demonstrate that the NP is nontoxic to non-cancerous cells.

8.      Why is the number of mice different across the groups?

9.      The authors should provide a survival curve if there was euthanasia involved.

10.  Figure 3, Y -axis says (% control), what does this mean?

11.  The authors should provide an IC50 curve of the free vs encapsulated drug (they mention it in line 275 but Figure 3A does not have a log curve depicting the IC50 changes). In line 280 the authors mention “Notably, using the trypan blue exclusion assay, the IC50 of ST1926 and ST1926-NPs was determined to be approximately 0.1 µM “, this indicates there is no advantage of using NP encapsulated drugs as the value does not change. Surprisingly, they use 4mg/kg in animals and show an efficacy of NP formulated drugs performing better than the free drug. What is the rationale behind this?

12.  In L 388 the authors mention “As expected ST1926 at 4 mg/kg did not reduce the tumor volumes (Figure 5A)”, why is this “as expected”. The use of NPs is usually to offset the limitations of small molecular drugs which include rapid clearance, off target toxicity, tolerance build up among others, the NPs themselves should not or usually do not provide any therapeutic efficacy. However, the high loading content ensures that repeated dosing can be avoided. In this case the authors show no difference in IC50 values between the free and NP encapsulated drug, so why and how do they come to this inference/conclusion of the NP+drug performing better than the free drug?

13.  Do the authors have any IHC data of the liver, kidney and lungs?

14.  Overall English must be improved. (L 394 we have to terminate the experiment for human reasons.” I believe this should be “humane”)

15.  The following references must be added:

 https://pubmed.ncbi.nlm.nih.gov/22349241/

https://www.ncbi.nlm.nih.gov/pmc/articles/PMC5619175/

https://pubmed.ncbi.nlm.nih.gov/31183964/

Comments on the Quality of English Language

English language, grammar and spell check needed

Author Response

Reviewer # 1

The authors here report the use of NP encapsulated adamantyl retinoid ST1926 for colorectal cancer therapy. My comments are as follows:

 We thank the reviewer for all the comments and suggestions.

  1. Schematic representation must be provided as Figure 1.

 We included a schematic representation for flash nanoprecipitation as Figure 1B.

As for the graphical abstract, we have added it separate from the rest of the manuscript figures as suggested by the journal recommendations.

  1. The methods must be described in detail (for example, the MWCO for the filters used in NP formulation, Cell source, passage number, age of mice etc.)

The molecular weight cut off (MWCO) of the filters is 50 KD. The human cell line: HCT116 was obtained from American Tissue Culture Collection (ATCC). Cell passages were between 18 and 22. Age of mice was between 4-6 weeks. These details were already added to the methods section, but we incorporated the passaging number of cells as highlighted in red (under cell culture in Methods section).

  1. Key data like encapsulation efficiency of the drug and its loading content are missing. The DLS plots based on intensity and number should be provided. Zeta potential is a key factor in NP stability, this data is missing.

We thank the reviewer for this insightful suggestion. We have added the drug loading efficiency and drug loading content of ST1926 nanoparticles (NPs) to the results section (Figure 2B). We also included the intensity size measurement to the supplementary section (Figure S4).

As for the zeta potential, our formulations use amphiphilic stabilizers which possess both hydrophobic and hydrophilic parts. Hence, the resulting NPs are not charged. Moreover, amphiphilic stabilizers provide steric stabilization, preventing particles from aggregating due to physical barriers rather than electrostatic repulsion. This is supported by the principle of flash nanoprecipitation, where steric stabilization provided by amphiphilic stabilizers ensures particle stability without relying on surface charge. Therefore, zeta potential measurements are not relevant in our study (Johnson and Prud'homme 2003).

  1. What is the critical aggregation constant (CAC) of the NPs?

The critical aggregation constant (CAC) can vary significantly depending on the solute and the formulation components. In the context of flash nanoprecipitation, the process rapidly moves the stabilizers from a soluble state to an aggregated state through mixing of solvents and non-solvents. This results in aggregation conditions that are significantly higher than the critical micelle concentrations. While CAC provides fundamental information on the thermodynamics of aggregation, determining it is beyond the scope of this work. Stabilizers aggregation studies for polystyrene-poly ethylene oxide (PSPEO) polymers in mixed solvents under conditions similar to this study can be found in this reference (Saad 2007).

 5. The authors should provide a drug release plot of the NPs.

We appreciate the reviewer’s comment. The highlight of this study is to understand the impact of the nanoparticle formulation under in vivo conditions. We conducted cell culture testing to establish the activity of the drug formulation on colorectal cancer cells. We have previously shown that ST1926 is potent in colorectal cancer in vitro models (Abdel-Samad, Aouad et al. 2018). Here, we aimed to verify that the nanoparticle formulation did not reduce the potent activity of ST1926 in vitro. It is important to note that drug release rates are not always representative of in vivo drug behavior.

  1. Do the authors have TEM images? SEM images are not very clear and since the surface morphology is not being investigated, it is more useful to have TEM images as well as a histogram representing the size distribution.

We thank the reviewer for the insightful comment. Unfortunately, the American University of Beirut does not have access to a TEM machine. Therefore, we used SEM to examine our NPs. We recognize that TEM could provide more extensive size distribution analysis and higher resolution images, but given our current resources, SEM is the best available option. The SEM instrument we are using provides high-resolution (up to 1.2 nm at 30 kV) imaging suitable for our needs.

  1. The authors have tested only one cell line, this is insufficient to draw conclusions. They should test their formulations and relevant controls on a non-cancerous cell line and demonstrate that the NP is nontoxic to non-cancerous cells.

We thank the reviewer for the perceptive comments. We have been previously published the effect of ST1926 on several human colorectal cancer cell lines with different genetic backgrounds and aggressiveness (HCT116, HCT116 p53-/-, HCT11621-/-, LoVo) (Abdel-Samad, Aouad et al. 2018). In the current study, we selected the HCT116 cell line to specifically test the effect of the nanoparticle formulation as it is a widely used colorectal cancer cell line in colorectal cancer studies. We have added this information in our revised manuscript.

We acknowledge the importance of evaluating the toxicity of our formulation on non-cancerous cells. As suggested, we tested the effect of ST1926 NPs on a normal human colon-like epithelial cell line: NCM460 by performing the MTT assay. ST1926-NPs were less sensitive on the normal-like cells and 0.5 µM concentrations did not affect their growth up to 72 hours post-treatment (Figure S7). However, ST1926-NPs at 0.5 µM resulted in approximately 60% growth inhibition in colorectal cells as early as 24 hours post- treatment (Figure 3A). We have added these news results to the text and as supplementary Figure S7.

  1. Why is the number of mice different across the groups?

We thank the reviewer for this comment. In vivo studies are subject to a significant degree of variability. To mitigate these variabilities and enhance the robustness of our data, we increased the number of mice in the key experimental group. This allows for a more statistically significant evaluation of the effect of NPs when comparing the novel formulation to the naked drug. It is worth noting that no animals used in the experiments were omitted. All mice were distributed into the groups in a manner to have conclusive results regarding the effect of our nanoparticle formulation.

  1. The authors should provide a survival curve if there was euthanasia involved.

We thank the reviewer for this comment. For humane reasons, the experiment was terminated when tumor volumes in the control groups hindered the mice mobility and their ability to feed on their own. As a result, we were not able to provide a survival curve.

  1. Figure 3, Y -axis says (% control), what does this mean?

The data are expressed as a percentage of the control group set as 100% (% control). The figure legend was adjusted accordingly in red.

  1. The authors should provide an IC50 curve of the free vs encapsulated drug (they mention it in line 275 but Figure 3A does not have a log curve depicting the IC50 changes). In line 280 the authors mention “Notably, using the trypan blue exclusion assay, the IC50 of ST1926 and ST1926-NPs was determined to be approximately 0.1 µM “, this indicates there is no advantage of using NP encapsulated drugs as the value does not change. Surprisingly, they use 4mg/kg in animals and show an efficacy of NP formulated drugs performing better than the free drug. What is the rationale behind this?

We thank the reviewer for the insightful comment. We provided an IC50 curve for the naked ST1926 and ST1926-NPs in supplementary Figure S6. The in vitro tests were designed to demonstrate that the formulation of ST1926 in NPs did not diminish its efficacy. Achieving similar results compared to the naked St1926 indicated that our formulation maintained the therapeutic potency of ST1926 in colorectal cancer cells.

Interestingly, formulating ST1926 in NPs enhanced the therapeutic efficiency of ST1926 in mice (Figure 5A) and increased the concentrations of ST1926 in tumors (new results in Table 2).

  1. In L 388 the authors mention “As expected ST1926 at 4 mg/kg did not reduce the tumor volumes (Figure 5A)”, why is this “as expected”. The use of NPs is usually to offset the limitations of small molecular drugs which include rapid clearance, off target toxicity, tolerance build up among others, the NPs themselves should not or usually do not provide any therapeutic efficacy. However, the high loading content ensures that repeated dosing can be avoided. In this case the authors show no difference in IC50 values between the free and NP encapsulated drug, so why and how do they come to this inference/conclusion of the NP+ drug performing better than the free drug?

We and others have previously published that the effective concentrations of ST1926 across different solid and liquid tumors ranged from 15 mg/kg (Nasr, Hmadi et al. 2015) to 50 mg/kg (Garattini, Parrella et al. 2004). ST1926 at 4 mg/kg body weight was not expected to reduce the tumor volumes. In fact, 4 mg/kg concentrations have not been previously studied for their efficacy in inhibiting tumor growth.

Consequently, despite the lack of difference between the IC50 values of free ST1926 and ST1926-NP formulation, ST1926-NPs showed a superior effect in comparison to free ST1926 in vivo. This enhanced effect could be attributed to the enhanced permeability and retention effect, which allows NPs to accumulate more effectively in tumor tissues which is indeed supported by our new data (Table 2).

  1. Do the authors have any IHC data of the liver, kidney and lungs?

Vital internal organs (liver, lungs, spleen, kidneys, intestines) were visually inspected and appeared normal compared to the control groups. However, none of the organs were collected for immunohistochemistry analysis as we did not observe any effects of the treatments on body weight and any other signs of visual toxicity such as changes in mice fur, eyes, and movement.

  1. Overall English must be improved. (L 394 we have to terminate the experiment for human reasons.” I believe this should be “humane”)

We thank the reviewer for the suggestion. We have included humane in text. Furthermore, Dr. Marwan El-Sabban kindly edited our manuscript.

  1. The following references must be added:

We thank the reviewer for the suggestions.

We added the following references.

https://pubmed.ncbi.nlm.nih.gov/22349241/

https://www.ncbi.nlm.nih.gov/pmc/articles/PMC5619175/

https://pubmed.ncbi.nlm.nih.gov/31183964/

Comments on the Quality of English Language

English language, grammar and spell check needed

We have edited and spell checked the manuscript.

Reviewer 2 Report

Comments and Suggestions for Authors

Assi et al described the nanoparticle drug with ST1926 and tested its efficacy to colorectal cancer. ST1926 NP did not give the change of concentration in plasma, however, the authors found that the tumor volume was more decreased by ST1926 NP than naked ST1926. They included it in discussion. The manuscript is well described and the contents are clear. I don’ t have so much things to be raised.

Minor points

Figure 1. It is better to describe the organization of nanoparticle by cartoons. 

Figure 2A. figure legend. I think PDI) is typo. Also better to describe the temperature to be incubated for 24h.

Figure 3. Figures of ST1926 and Figures of ST1926 NP were not really separated. Better to have some distance between the graph.

Author Response

Reviewer # 2

Assi et al described the nanoparticle drug with ST1926 and tested its efficacy to colorectal cancer. ST1926 NP did not give the change of concentration in plasma, however, the authors found that the tumor volume was more decreased by ST1926 NP than naked ST1926. They included it in discussion. The manuscript is well described and the contents are clear. I don’ t have so much things to be raised.
We thank the reviewer for all the comments.

Minor points
1. Figure 1. It is better to describe the organization of nanoparticle by cartoons.
We appreciate the reviewer’s comment. We adjusted Figure 1 to include a schematic of our nanoparticle formulation (Figure 1C).

2. Figure 2A. figure legend. I think PDI) is typo. Also better to describe the temperature to be incubated for 24h.
We thank the reviewer for this comment. We adjusted the figure legend in figure 2A and included the temperature of incubation in the figure legend (Figure 2A) and in the methods section under characterization of ST1926-nanoparticles.

3. Figure 3. Figures of ST1926 and Figures of ST1926 NP were not really separated. Better to have some distance between the graph.
We thank the reviewer for this comment. We adjusted Figure 3 accordingly.

Reviewer 3 Report

Comments and Suggestions for Authors

This work prepared ST1926 nanoparticle for treatment of colorectal cancer. Although flash nanoprecipitation was used to prepare the nanoparticles, there are many issues for this work.

(1)   What is the uniqueness of the flash nanoprecipitation? I didn’t found any differences of the prepared nanoparticles from the ones by the other methods.

(2)   The drug release curves should be added.

(3)   Why is the cholesterol needed for this formulation? What are the main roles of cholesterol. How is the nanoparticle if there is no cholesterol.

(4)   The in vivo distribution of nanoparticles should be investigated.

(5)   The drug concentrations in the different organs or tissues should be investigated and compared with the pure ST1926.

Comments on the Quality of English Language

No comments

Author Response

Reviewer # 3

This work prepared ST1926 nanoparticle for treatment of colorectal cancer. Although flash nanoprecipitation was used to prepare the nanoparticles, there are many issues for this work.
We appreciate the reviewer’s thoughtful comments and valuable suggestions.

1. What is the uniqueness of the flash nanoprecipitation? I didn’t found any differences of the prepared nanoparticles from the ones by the other methods.
We thank the reviewer for this comment.
Flash nanoprecipitation (FNP) has several advantages compared to other methods, including continuous processing, ease of scale up, and the ability to accommodate multiple active components in nanoparticles. In addition, from a pharmaceutical manufacturing perspective, FNP presents a versatile technology for commercial-scale production of nanoparticle-based formulations (Saad and Prud’homme 2016). This information has been added in the introduction.
The FNP technology has been instrumental in the mass production of COVID-19 vaccines. It was used to efficiently encapsulate mRNA in lipid nanoparticles, ensuring stability and effective delivery of the vaccine’s genetic material into cell (Lyon 2022).

2. The drug release curves should be added.
We appreciate the reviewer’s comment.
The nanoparticle formulation developed via FNP results in precipitated drug stabilized by amphiphilic molecules at the surface that provide steric hindrance and minimizes further aggregation of the drug. Hence, there is no drug reservoir as in liposomes or other encapsulation forms to justify the investigation of drug release from the encapsulating particles.
The highlight of this study is to understand the impact of the nanoparticle formulation under in vivo conditions. We conducted cell culture testing to establish the activity of the drug formulation on colorectal cancer cells. We have previously shown that ST1926 is potent in colorectal cancer in vitro models (Abdel-Samad, Aouad et al. 2018). Here, we aimed to verify that the nanoparticle formulation did not reduce the potent activity of ST1926 in vitro. It is important to note that drug release rates are not always representative of in vivo drug behavior, particularly for non-oral drug products (Shen and Burgess 2015).

3. Why is the cholesterol needed for this formulation? What are the main roles of cholesterol. How is the nanoparticle if there is no cholesterol.
We are grateful for this comment.
We incorporated cholesterol as a co-stabilizer into our nanoparticle formulation based on insights elucidated by Wan et al (Wan, Wong et al. 2019). Cholesterol, known for its role as a hydrophobic stabilizer, has demonstrated its ability to improve the stability of various nanoparticle formulations. This enhancement is achieved by facilitating the alignment and orientation of amphiphilic stabilizers during storage. Moreover, drugs within the ACD/Log P
range of 2 to 9 often exhibit instability attributed to Ostwald ripening and non-equilibrium molecular orientation of amphiphilic stabilizers. The incorporation of cholesterol as a co-stabilizer addresses this critical limitation of flash nanoprecipitation, considering that the majority of drugs fall within this ACD/Log P range (Wan, Wong et al. 2019). By postulating that the presence of a hydrophobic co-stabilizer facilitates the formation of a mixed particle core with the drug; while also exposing some of the co-stabilizer on the core surface, we anticipated improved interactions between the hydrophobic particle core and the hydrophobic tails of the amphiphilic stabilizer. This strategic utilization of cholesterol highlights its potential to mitigate instability issues associated with flash nanoprecipitation-produced drug nanosuspensions, thus expanding the applicability and efficacy of such formulations in pharmaceutical applications. Given that the ACD/log P of ST1926 is 6.74, indicating its hydrophobic nature, we examined that cholesterol reduced particle size and enhanced stability.
We have highlighted this information in the Results section.

4. The in vivo distribution of nanoparticles should be investigated.
We thank the reviewer for this insightful comment.
We have performed new studies measuring the concentrations of ST1926 in tumor samples treated with naked ST1926 and ST1926-NPs. ST1926 concentrations in the tumors were analyzed by HPLC-QTOF/MS and showed that ST1926 was detected in 6 out of 10 tumors treated with ST1926-NP, with an average concentration of 44.4 ng/ml (Table 2). In contrast, no ST1926 was detected in tumor samples treated with ST1926 alone, as it was below the limit of detection (LOD) (1 ng/ml) (Table 2).
We have added this new information in the Methods and Results sections.

5. The drug concentrations in the different organs or tissues should be investigated and compared with the pure ST1926.
We thank the reviewer for this perceptive comment.
The highlight of our study was to determine the efficacy of our nanoparticle formulation in comparison to the naked drug on tumor volumes and drug concentrations. Future experiments will focus on the mechanistic studies, pharmacokinetic characterization, and tissue distribution analysis to thoroughly assess the pharmacological characteristics of ST1926 nanoparticles and enhance its therapeutic efficacy in colorectal cancer treatment.

Round 2

Reviewer 1 Report

Comments and Suggestions for Authors

The authors have addressed every comment made to them to the best of their ability and resources available. I recommend this for publication

Author Response

Reviewer # 1

The authors here report the use of NP encapsulated adamantyl retinoid ST1926 for colorectal cancer therapy. My comments are as follows:

We thank the reviewer for all the comments and suggestions.

  1. Schematic representation must be provided as Figure 1.

We included a schematic representation for flash nanoprecipitation as Figure 1B.

As for the graphical abstract, we have added it separate from the rest of the manuscript figures as suggested by the journal recommendations.

  1. The methods must be described in detail (for example, the MWCO for the filters used in NP formulation, Cell source, passage number, age of mice etc.)

The molecular weight cut off (MWCO) of the filters is 50 KD. The human cell line: HCT116 was obtained from American Tissue Culture Collection (ATCC). Cell passages were between 18 and 22. Age of mice was between 4-6 weeks. These details were already added to the methods section, but we incorporated the passaging number of cells as highlighted in red (under cell culture in Methods section).

  1. Key data like encapsulation efficiency of the drug and its loading content are missing. The DLS plots based on intensity and number should be provided. Zeta potential is a key factor in NP stability, this data is missing.

We thank the reviewer for this insightful suggestion.

We have added the drug loading efficiency and drug loading content of ST1926 nanoparticles (NPs) to the results section (Figure 2B). We also included the intensity size measurement to the supplementary section (Figure S4).

As for the zeta potential, our formulations use amphiphilic stabilizers which possess both hydrophobic and hydrophilic parts. Hence, the resulting NPs are not charged. Moreover, amphiphilic stabilizers provide steric stabilization, preventing particles from aggregating due to physical barriers rather than electrostatic repulsion. This is supported by the principle of flash nanoprecipitation, where steric stabilization provided by amphiphilic stabilizers ensures particle stability without relying on surface charge. Therefore, zeta potential measurements are not relevant in our study (Johnson and Prud'homme 2003).

  1. What is the critical aggregation constant (CAC) of the NPs?

The critical aggregation constant (CAC) can vary significantly depending on the solute and the formulation components. In the context of flash nanoprecipitation, the process rapidly moves the stabilizers from a soluble state to an aggregated state through mixing of solvents and non-solvents. This results in aggregation conditions that are significantly higher than the critical micelle concentrations. While CAC provides fundamental information on the thermodynamics of aggregation, determining it is beyond the scope of this work. Stabilizers aggregation studies for polystyrene-poly ethylene oxide (PSPEO) polymers in mixed solvents under conditions similar to this study can be found in this reference (Saad 2007).

  1. The authors should provide a drug release plot of the NPs.

We appreciate the reviewer’s comment.

The nanoparticle formulation developed via FNP results in precipitated drug stabilized by amphiphilic molecules at the surface that provide steric hindrance and minimizes further aggregation of the drug. Hence, there is no drug reservoir as in liposomes or other encapsulation forms to justify the investigation of drug release from the encapsulating particles.

The highlight of this study is to understand the impact of the nanoparticle formulation under in vivo conditions. We conducted cell culture testing to establish the activity of the drug formulation on colorectal cancer cells. We have previously shown that ST1926 is potent in colorectal cancer in vitro models (Abdel-Samad, Aouad et al. 2018). Here, we aimed to verify that the nanoparticle formulation did not reduce the potent activity of ST1926 in vitro. It is important to note that drug release rates are not always representative of in vivo drug behavior, particularly for non-oral drug products (Shen and Burgess 2015).

  1. Do the authors have TEM images? SEM images are not very clear and since the surface morphology is not being investigated, it is more useful to have TEM images as well as a histogram representing the size distribution.

We thank the reviewer for the insightful comment.

Unfortunately, the American University of Beirut does not have access to a TEM machine. Therefore, we used SEM to examine our NPs. We recognize that TEM could provide more extensive size distribution analysis and higher resolution images, but given our current resources, SEM is the best available option. The SEM instrument we are using provides high-resolution (up to 1.2 nm at 30 kV) imaging suitable for our needs.

  1. The authors have tested only one cell line, this is insufficient to draw conclusions. They should test their formulations and relevant controls on a non-cancerous cell line and demonstrate that the NP is nontoxic to non-cancerous cells.

We thank you the reviewer for the perceptive comments.

We have previously published the effect of ST1926 on several human colorectal cancer cell lines with different genetic backgrounds and aggressiveness (HCT116, HCT116 p53-/-, HCT11621-/-, LoVo) (Abdel-Samad, Aouad et al. 2018). In the current study, we selected the HCT116 cell line to specifically test the effect of the nanoparticle formulation as it is a widely used colorectal cancer cell line in colorectal cancer studies. We have added this information in our revised manuscript.

We acknowledge the importance of evaluating the toxicity of our formulation on non-cancerous cells. As suggested, we tested the effect of ST1926 NPs on a normal human colon-like epithelial cell line: NCM460 by performing the MTT assay. ST1926-NPs were less sensitive on the normal-like cells and 0.5 µM concentrations did not affect their growth up to 72 hours post-treatment (Figure S7). However, ST1926-NPs at 0.5 µM resulted in approximately 60% growth inhibition in colorectal cells as early as 24 hours post- treatment (Figure 3A). We have added these news results to the text and as supplementary Figure S7.

  1. Why is the number of mice different across the groups?

We thank the reviewer for this comment.

In vivo studies are subject to a significant degree of variability. To mitigate these variabilities and enhance the robustness of our data, we increased the number of mice in the key experimental group. This allows for a more statistically significant evaluation of the effect of NPs when comparing the novel formulation to the naked drug. It is worth noting that no animals used in the experiments were omitted. All mice were distributed into the groups in a manner to have conclusive results regarding the effect of our nanoparticle formulation.

  1. The authors should provide a survival curve if there was euthanasia involved.

We thank the reviewer for this comment.

For humane reasons, the experiment was terminated when tumor volumes in the control groups hindered the mice mobility and their ability to feed on their own. As a result, we were not able to provide a survival curve. It is important to note that all animals survived until the date of euthanasia.

  1. Figure 3, Y -axis says (% control), what does this mean?

The data are expressed as a percentage of the control group set as 100% (% control). The figure legend was adjusted accordingly in red.

  1. The authors should provide an IC50 curve of the free vs encapsulated drug (they mention it in line 275 but Figure 3A does not have a log curve depicting the IC50 changes). In line 280 the authors mention “Notably, using the trypan blue exclusion assay, the IC50 of ST1926 and ST1926-NPs was determined to be approximately 0.1 µM “, this indicates there is no advantage of using NP encapsulated drugs as the value does not change. Surprisingly, they use 4mg/kg in animals and show an efficacy of NP formulated drugs performing better than the free drug. What is the rationale behind this?

We thank the reviewer for the insightful comment.

We provided an IC50 curve for the naked ST1926 and ST1926-NPs in supplementary Figure S6. The in vitro tests were designed to demonstrate that the formulation of ST1926 in NPs did not diminish its efficacy. Achieving similar results compared to the naked St1926 indicated that our formulation maintained the therapeutic potency of ST1926 in colorectal cancer cells.

Interestingly, formulating ST1926 in NPs enhanced the therapeutic efficiency of ST1926 in mice (Figure 5A) and increased the concentrations of ST1926 in tumors (new results in Table 2).

  1. In L 388 the authors mention “As expected ST1926 at 4 mg/kg did not reduce the tumor volumes (Figure 5A)”, why is this “as expected”. The use of NPs is usually to offset the limitations of small molecular drugs which include rapid clearance, off target toxicity, tolerance build up among others, the NPs themselves should not or usually do not provide any therapeutic efficacy. However, the high loading content ensures that repeated dosing can be avoided. In this case the authors show no difference in IC50 values between the free and NP encapsulated drug, so why and how do they come to this inference/conclusion of the NP+ drug performing better than the free drug?

We and others have previously published that the effective concentrations of ST1926 across different solid and liquid tumors ranged from 15 mg/kg (Nasr, Hmadi et al. 2015) to 50 mg/kg (Garattini, Parrella et al. 2004). ST1926 at 4 mg/kg body weight was not expected to reduce the tumor volumes. In fact, 4 mg/kg concentrations have not been previously studied for their efficacy in inhibiting tumor growth.

Consequently, despite the lack of difference between the IC50 values of free ST1926 and ST1926-NP formulation, ST1926-NPs showed a superior effect in comparison to free ST1926 in vivo. This enhanced effect could be attributed to the enhanced permeability and retention effect, which allows NPs to accumulate more effectively in tumor tissues which is indeed supported by our new data (Table 2).

  1. Do the authors have any IHC data of the liver, kidney and lungs?

Vital internal organs (liver, lungs, spleen, kidneys, intestines) were visually inspected and appeared normal compared to the control groups. However, none of the organs were collected for immunohistochemistry analysis as we did not observe any effects of the treatments on body weight and any other signs of visual toxicity such as changes in mice fur, eyes, and movement.

  1. Overall English must be improved. (L 394 we have to terminate the experiment for human reasons.” I believe this should be “humane”)

We thank the reviewer for the suggestion.

We have included humane in text. Furthermore, Dr. Marwan El-Sabban kindly edited our manuscript.

  1. The following references must be added:

We thank the reviewer for the suggestions.

We added the following references.

https://pubmed.ncbi.nlm.nih.gov/22349241/

https://www.ncbi.nlm.nih.gov/pmc/articles/PMC5619175/

https://pubmed.ncbi.nlm.nih.gov/31183964/

Comments on the Quality of English Language

English language, grammar and spell check needed

We have edited and spell checked the manuscript.